# The link between liver fat and cardiometabolic diseases is highlighted by genome-wide association study of MRI-derived measures of body composition

Dennis van der Meer [1,2✉], Tiril P. Gurholt [1], Ida E. Sønderby [1,3,4], Alexey A. Shadrin[1], Guy Hindley [1,5], Zillur Rahman[1], Ann-Marie G. de Lange[1,6,7], Oleksandr Frei[1,8], Olof D. Leinhard[9,10,11], Jennifer Linge[9,10], Rozalyn Simon[10,11], Dani Beck[1,12,13], Lars T. Westlye [1,4,12], Sigrun Halvorsen [14], Anders M. Dale [15], Tom H. Karlsen [16,17], Tobias Kaufmann [1,18] & Ole A. Andreassen [1,4]

Obesity and associated morbidities, metabolic associated fatty liver disease (MAFLD) included, constitute some of the largest public health threats worldwide. Body composition and related risk factors are known to be heritable and identification of their genetic determinants may aid in the development of better prevention and treatment strategies. Recently, large-scale whole-body MRI data has become available, providing more specific measures of body composition than anthropometrics such as body mass index. Here, we aimed to elucidate the genetic architecture of body composition, by conducting genome-wide association studies (GWAS) of these MRI-derived measures. We ran both univariate and multivariate GWAS on fourteen MRI-derived measurements of adipose and muscle tissue distribution, derived from scans from 33,588 White European UK Biobank participants (mean age of 64.5 years, 51.4% female). Through multivariate analysis, we discovered 100 loci with distributed effects across the body composition measures and 241 significant genes primarily involved in immune system functioning. Liver fat stood out, with a highly discoverable and oligogenic architecture and the strongest genetic associations. Comparison with 21 common cardiometabolic traits revealed both shared and specific genetic influences, with higher mean heritability for the MRI measures ($h^2 = .25$ vs. .13, $p = 1.8 \times 10^{-7}$). We found substantial genetic correlations between the body composition measures and a range of cardiometabolic diseases, with the strongest correlation between liver fat and type 2 diabetes ($r_g = .49$, $p = 2.7 \times 10^{-22}$). These findings show that MRI-derived body composition measures complement conventional body anthropometrics and other biomarkers of cardiometabolic health, highlighting the central role of liver fat, and improving our knowledge of the genetic architecture of body composition and related diseases.

A full list of author affiliations appears at the end of the paper.

Obesity and associated cardiometabolic diseases are currently considered one of the largest global public health concerns[1,2]. Over one-third of the United States adult population qualifies for a diagnosis of metabolic syndrome[3], characterized by excessive visceral adiposity, insulin resistance, hypertension, low high-density lipoprotein cholesterol, and hypertriglyceridemia[4,5]. Metabolic syndrome substantially increases the risk of coronary artery disease, type 2 diabetes, cancer, and metabolic associated fatty liver disease (MAFLD, previously described as non-alcoholic fatty liver disease[6])[7–11]. As shown by our previous work, body composition is also strongly associated with brain structure and brain disorders, which are among the most costly and debilitating medical conditions in the world[12–14]. An improved understanding of the genetic and biological determinants of body composition is needed to provide insights into the complex interplay between metabolic factors, prevent and treat multiple highly prevalent conditions, and improve public health outcomes[2,10].

Body composition is partly determined by a complex constellation of interacting metabolic processes and inter-organ crosstalk that may become dysregulated and lead to metabolic syndrome[15]. In susceptible individuals, excessive energy intake, stored as visceral adipose tissue (VAT), combined with insulin resistance, leads to heightened lipolysis and release of free fatty acids[16]. Increased free fatty acid flux to the liver results in hypertriglyceridemia, which in turn contributes to dyslipidemia and atherosclerosis. Lipolysis in VAT further promotes insulin resistance and gluconeogenesis and increases pro-inflammatory reactions that exacerbate endothelial dysfunction and hypertension[16]. This is reflected in heightened levels of pro-inflammatory markers among individuals with metabolic syndrome[17]. Muscle mass is also a determinant of cardiometabolic health[18], as skeletal muscle constitutes the largest insulin-sensitive tissue in the body and is the primary site for insulin-stimulated glucose utilization[19]. Still, the nature and extent of overlap between these different determinants of cardiometabolic functioning remain unclear.

Measures of localized adipose tissue, liver fat and regional muscle volume can now be accurately extracted from whole-body MRI scans[20–23]. Body anthropometrics such as waist circumference and body mass index (BMI) lack a direct connection to pathophysiology[5,24] Measures of regional adipose tissue, most accurately and comprehensively identified through MRI[25,26], may offer sensitive proxies of cardiometabolic health and therefore complement these common measures[27]. This is further suggested by research indicating they have independent associations with cardiometabolic diseases and improve risk prediction beyond body anthropometrics[28–30].

In addition to social and physical environmental factors[31], genetically determined individual differences play a substantial role in regulating body composition[32–34]. Cardiometabolic risk factors have both unique and shared genetic correlates[35]. Much less is known about the genetics of specific MRI-derived body composition measures. We aimed to map the unique and shared genetic architectures across the MRI-derived body composition to provide a holistic understanding of the interplay between different tissue types and their role in metabolic syndrome and cardiometabolic health. We further sought to identify the extent of genetic overlap between these measures and common medical conditions, as such information promotes research into shared molecular pathways and therefore a better understanding of the underlying biology.

## Results

We conducted GWASs of fourteen MRI-derived muscle and adipose tissue distribution measures and investigated the genetic

**Table 1 MRI-derived measures of body composition included in this study.**

| Measure | Abbreviation | N | # loci |
|---|---|---|---|
| Abdominal subcutaneous adipose tissue | ASAT | 33,532 | 1 |
| Visceral adipose tissue | VAT | 33,542 | 2 |
| Anterior thigh muscle volume | ATMV | 32,978 | 8 |
| Posterior thigh muscle volume | PTMV | 33,022 | 9 |
| Anterior thigh muscle fat infiltration (%) | ATMFI | 32,911 | 18 |
| Posterior thigh muscle fat infiltration (%) | PTMFI | 32,956 | 25 |
| Weight-muscle-ratio | WMR | 32,970 | 1 |
| Abdominal fat ratio | AFR | 32,939 | 1 |
| Liver proton density fat fraction (%) | LPDFF | 33,235 | 8 |
| VAT/height$^2$ | VATi | 32,564 | 2 |
| ASAT/height$^2$ | ASATi | 32,573 | 1 |
| ATMV/height$^2$ | ATMVi | 32,017 | 2 |
| PTMV/height$^2$ | PTMVi | 32,059 | 0 |
| Total thigh muscle volume z-score | TTMVz | 31,977 | 4 |

Further provided are the available sample size and number of loci discovered through univariate GWAS.

link to conventional cardiometabolic risk factors. We included six measures of adipose tissue distribution: abdominal subcutaneous adipose tissue, VAT, abdominal fat ratio, anterior and posterior thigh muscle fat infiltration, and liver protein density fat fraction. Additionally, we investigated three measures related to thigh muscle tissue, namely anterior and posterior thigh muscle volume and weight-to-muscle ratio. We further analyzed visceral and abdominal adipose tissue, and anterior and posterior muscle volume, divided by standing height in meters squared, and total thigh muscle volume z-score (sex-, height-, weight-, and BMI-invariant)[36]. See Table 1 for an overview of these measures, and the Methods section for protocols and definitions. Given a total of fourteen individual measures, we set the univariate GWAS significance threshold at $\alpha = 5 \times 10^{-8}/14 = 3.6 \times 10^{-9}$. Our sample for the main analyses consisted of 33,588 unrelated White European participants of the UK Biobank (UKB), with a mean age of 64.5 years (standard deviation (SD) 7.5 years), 51.4% female. We pre-residualized all measures for age, sex, test center, genotyping array and the first twenty genetic principal components to control for population stratification[37].

**Univariate GWAS.** Univariate GWASs on the individual measures revealed a total of 82 loci, including 50 unique, surpassing the study-wide significance threshold of $3.6 \times 10^{-9}$. Two loci stood out with highly significant p-values, on chromosome 19 (lead rs58542926, $p = 4.4 \times 10^{-110}$) and chromosome 22 (lead rs738409, $p = 2.8 \times 10^{-161}$), both identified in the GWAS on liver fat. Using converging positional, eQTL, and chromatin interaction information (see Methods), we mapped these loci to genes previously coupled to MAFLD (rs738409: *PNPLA3, SAMM50, PARVB*)[38] as well as inflammatory processes and cancer (rs58542926: *TM6SF2, CD99*)[39]. Supplementary Fig. 1 contains Manhattan plots and Supplementary Data 1 lists overviews of all loci discovered together with mapped genes.

Additionally, we assessed the generalization of the discovered loci in an additional set of 5042 non-White European UKB participants with identical processing steps. Of the 79 lead single-nucleotide polymorphisms (SNPs) available in this set, 77 had effects in the same direction as the main analyses (97.5%, sign-test $p < 1 \times 10^{-16}$). Thus, our results suggest a cross-ethnicity generalization of these genetic associations with MRI-derived measures

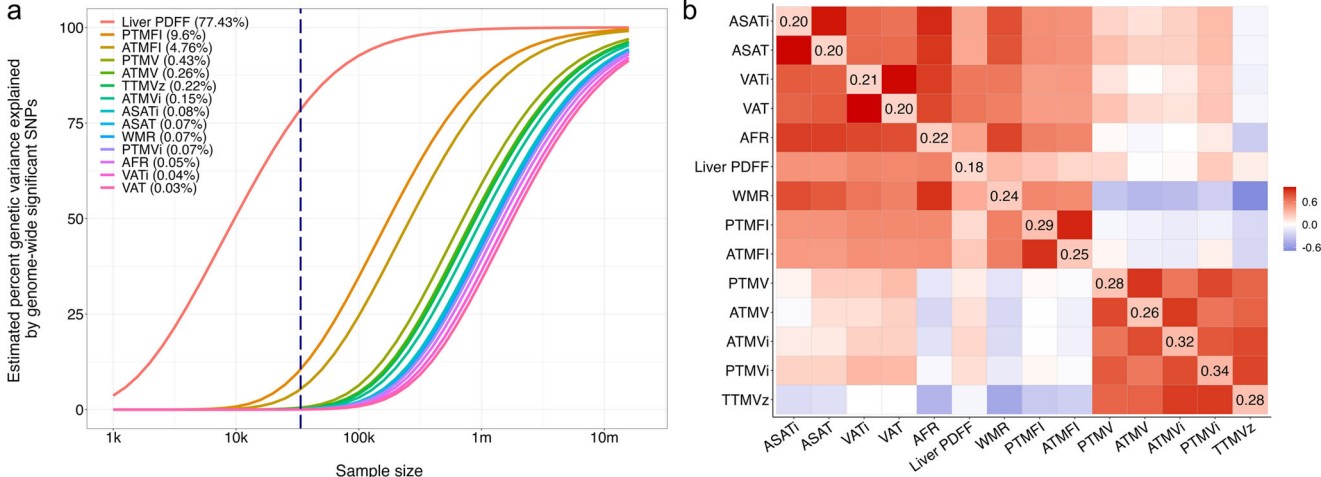

**Fig. 1 Comparison of the genetic architecture of individual body composition measures. a** The relation between genetic variance explained by genome-wide significant hits (y-axis) and sample size (x-axis) for each measure (solid-colored lines). The vertical dashed blue line marks the current sample size, with the corresponding percent genetic variance explained indicated between brackets in the legend. **b** Correlation between the measures, with phenotypic correlation shown in lower-left section and genetic correlation in the upper-right section, and heritability on the diagonal. Abbreviations: *ASAT* abdominal subcutaneous adipose tissue, *VAT* visceral adipose tissue, *AFR* abdominal fat ratio, *WMR* weight-muscle-ratio, *ATMV* anterior thigh muscle volume, *PTMV* posterior thigh muscle volume, *ATMFI* anterior thigh muscle fat infiltration, *PTMFI* posterior thigh muscle fat infiltration, *Liver PDFF* liver proton density fat fraction, *TTMVz* total thigh muscle volume z-score, *i* index, referring to a measure divided by standing height[2].

of body composition, despite the known high variability of body anthropometrics across ethnicities[5,24].

In total, we identified eight study-wide significant loci for liver fat, validating those found in a previous GWAS of organ tissue using different measurement protocols[40]. Gene-based analysis through Multi-marker Analysis of GenoMic Annotation (MAGMA) identified 31 genome-wide significant genes, including the three primary MAFLD genes (TM6SF2 $p = 1.7 \times 10^{-15}$, PNPLA3 $p = 7.8 \times 10^{-15}$, and TMC4-MBOAT7 $p = 7.2 \times 10^{-9}$)[41–44], further confirming the strong biological validity of this liver fat measure and its connection to MAFLD. Functional annotation of the set of 31 genes revealed differential expression in the liver, pancreas, and subcortical brain regions and significant enrichment among Gene Ontology (GO) biological processes specifically related to lipid homeostasis and metabolic processes. Supplementary Data 2 and 3 further contain results of gene set enrichment analyses for each individual measure.

Next, we estimated the polygenicity and effect size variance ('discoverability') by fitting a Gaussian mixture model of null and non-null effects to the GWAS summary statistics using MiXeR[45,46]. The results are summarized in Fig. 1a, depicting the estimated proportion of genetic variance explained by discovered SNPs for each measure as a function of sample size. This illustrates that body MRI measures generally show genetic architectures similar to e.g., brain MRI measures, characterized by high polygenicity[47,48]. However, the notable exception is liver fat, with substantially lower polygenicity and higher discoverability than the other measures, in line with the relatively few highly significant associations we identified through the GWAS.

Figure 1b visualizes the phenotypic and genetic correlations between each pair of measures, confirming a strong structure and a subdivision between adipose- and muscle-related measures. SNP-based heritability ranged from 18% to 34% (all $p < 1 \times 10^{-16}$); see the diagonal of Fig. 1b.

**Multivariate GWAS.** Gene variants are likely to have distributed effects across these measures of body composition, as they are correlated components of the same biological system. To identify variants influencing body composition as a whole, we also jointly analyzed all measures through the Multivariate Omnibus Statistical Test (MOSTest)[49], which increases statistical power in a scenario of shared genetic signal across the univariate measures[49–51]. After applying a rank-based inverse normal transformation, we performed MOSTest on the residualized measures, yielding a multivariate association with 9.1 million SNPs included.

MOSTest revealed 100 significant independent loci across all MRI-derived measures (Fig. 2a and Supplementary Data 4). Figure 2b visualizes the significance of the association between the individual measures and each of the 100 loci, illustrating the presence of many shared but also specific genetic variants.

MAGMA identified 241 significant genes after multiple comparison correction ($\alpha = 0.05/18,203$), with highly significant differential expression in the liver, pancreas, heart, muscle, and several other tissues (Fig. 3). Coupling the significant genes to the Reactome database[52] indicated most prominent associations with the adaptive immune system and cytokine signaling ($p < 1 \times 10^{-16}$), see Supplementary Data 5 and Supplementary Fig. 2 for an overview.

**Comparison of genetic architecture with cardiometabolic risk factors.** To establish whether the loci discovered through the univariate GWAS of the body MRI measures are novel compared to related measures of cardiometabolism, we additionally ran univariate GWAS on 21 secondary measures of anthropometric and cardiometabolic factors (e.g., BMI, triglycerides, cholesterol, blood pressure; see Table 2). To ascertain whether the body MRI measures truly allow for more discovery, without being confounded by differences in sample size, we restricted these analyses to the same sample of individuals with available MRI data ($N = 33,588$). These analyses showed that the large majority of loci were indeed novel; Supplementary Fig. 3 summarizes this, showing for each discovered variant whether it was whole-genome significant for each of the primary and secondary measures.

We further ran multivariate GWAS on this separate set of measures in the full UKB sample, consisting 377,950 unrelated White European UKB participants. Through MOSTest, we found

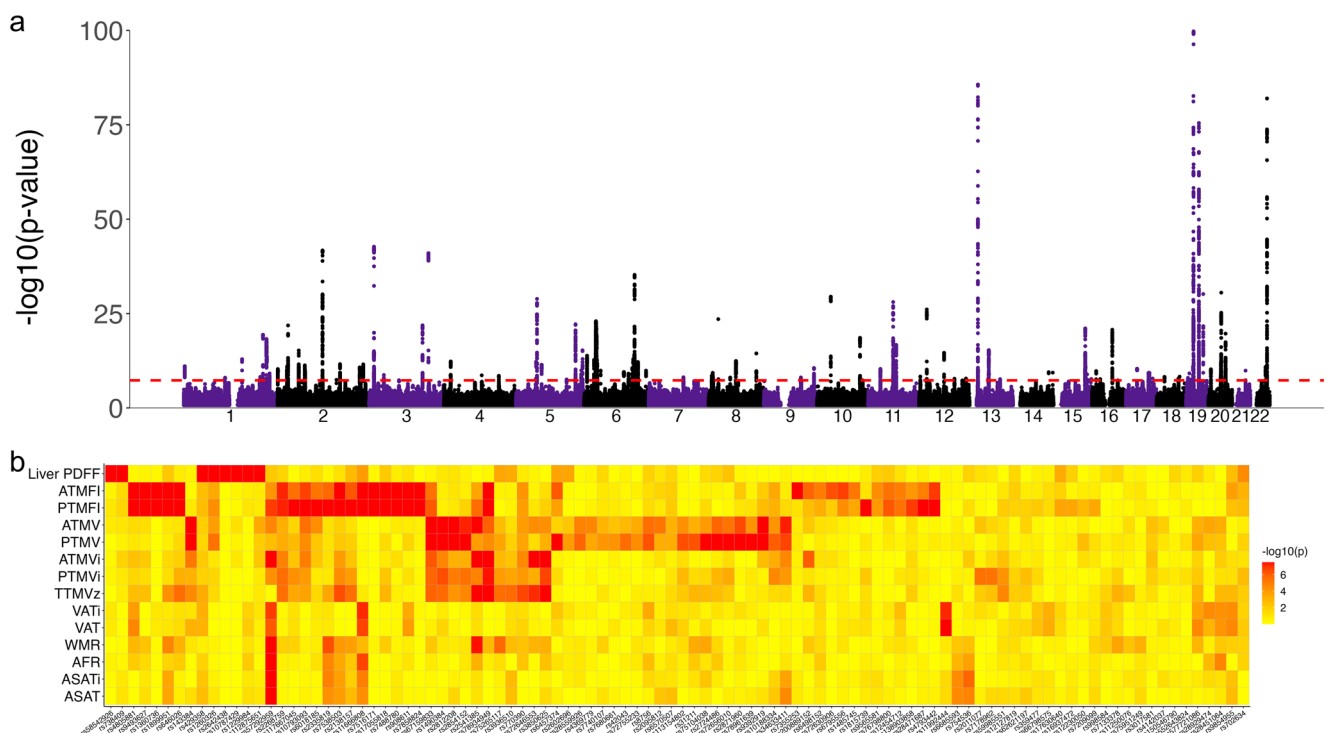

**Fig. 2 Multivariate locus discovery. a** Manhattan plot of the multivariate GWAS on all MRI-derived body composition measures, with the observed $-\log10(p)$ of each SNP shown on the y-axis. The x-axis shows the relative genomic location, grouped by chromosome, and the red dashed line indicates the whole-genome significance threshold of $5 \times 10^{-8}$. The y-axis is clipped at $-\log10(p) = 75$. **b** Heatmap showing $-\log10(p)$ of the association between the lead variants of MOSTest-identified independent loci (x-axis) and each of the individual MRI measures (y-axis). The values are capped at 7.5 ($p = 5 \times 10^{-8}$).

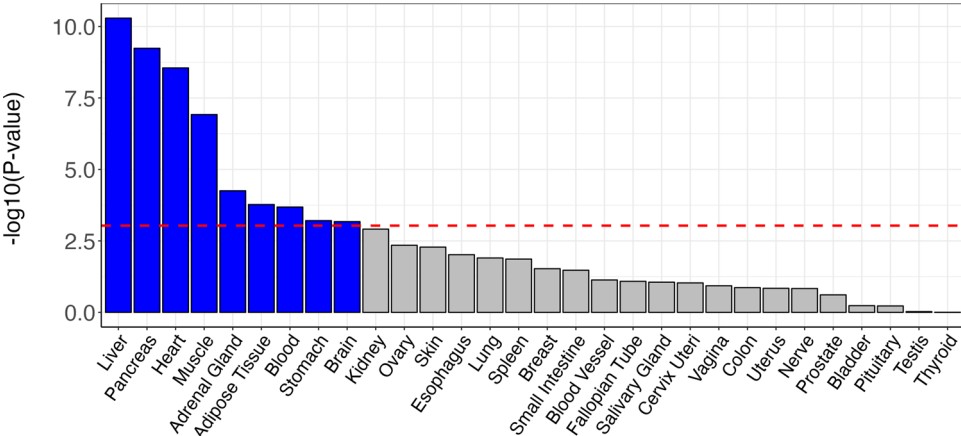

**Fig. 3 Tissue-specific differential expression of the set of significant genes identified through the multivariate GWAS on MRI-derived measures of body composition.** The red-dotted line indicates the multiple comparisons-corrected significance threshold.

1173 genome-wide significant loci with $\alpha = 5 \times 10^{-8}$ (list provided in Supplementary Data 6). Of the 100 loci identified through the primary multivariate analysis of MRI-derived body composition measures, 91 were significant in this secondary analysis in a larger sample. This indicates that, combined, these sets of measures overall are influenced by the same network of biological processes.

The heritability of the MRI-derived measures (mean $h^2 = 0.25$, 95% CI [0.22, 0.28]) was significantly higher than the body anthropometrics and other biomarkers (mean $h^2 = 0.13$, 95% CI [0.10, 0.15]), $p = 1.8 \times 10^{-7}$. The heritability per measure is further provided in Supplementary Fig. 4. As shown in Fig. 4, these measures generally showed higher genetic correlations with

the MRI-derived measures of adipose tissue than the muscle-related measures. Further, BMI, hip/waist circumference, and waist-to-hip-ratio were genetically correlated with nearly all body MRI measures.

**Genetic correlation with cardiovascular, metabolic and mental disorders.** Next, we analyzed the genetic overlap of the MRI-derived measures with medical conditions previously linked to cardiometabolic health, selecting recent GWAS with adequate power[53–59]. As shown in Fig. 5a, the strongest association across all measures was found for liver fat, with a genetic correlation of 0.49 ($p = 2.7 \times 10^{-22}$) with type 2 diabetes. Coronary artery

**Table 2 Measures of cardiometabolic health used in the secondary analyses, together with abbreviations and maximum available sample sizes.**

| Measure | Abbreviation | N |
|---|---|---|
| Cholesterol | | 3,60,319 |
| High-density lipoproteins | HDL | 3,29,791 |
| Low-density lipoproteins | LDL | 3,59,650 |
| Triglycerides | | 3,60,036 |
| Apolipoprotein A | | 3,27,953 |
| Apolipoprotein B | | 3,58,576 |
| Cholesterol to HDL | | 3,29,727 |
| ApoB to ApoA | | 3,26,320 |
| C-reactive protein | CRP | 3,59,539 |
| Glucose | | 3,29,565 |
| Glycated hemoglobin | HbA1c | 3,60,260 |
| Alanine aminotransferase | ALT | 3,60,190 |
| Aspartate aminotransferase | AST | 3,59,000 |
| Gamma-glutamyl transferase | GGT | 3,60,136 |
| Creatinine | | 3,60,141 |
| Body mass index | BMI | 3,76,747 |
| Waist circumference | | 3,77,321 |
| Hip circumference | | 3,77,283 |
| Waist-to-hip ratio | WHR | 3,77,249 |
| Diastolic blood pressure | DBP | 33,784 |
| Systolic blood pressure | SBP | 33,784 |

disease was found to have highly significant positive genetic correlations with visceral and subcutaneous adipose tissue. Overall, we found weak negative genetic correlations with muscle tissue measures and stronger positive genetic correlations with adipose tissue measures, with two exceptions; anorexia nervosa showed the opposite direction of correlation compared to the other conditions, and there was no discernible pattern for schizophrenia. Genetic correlations with the anthropometric and metabolic measures are provided in Fig. 5b for comparison, indicating that the adipose tissue measures are as strong as or stronger correlated with these conditions than the conventional body anthropometrics.

**Sex-specific analyses**. Given that the body composition of men and women differs substantially, we provide sex-stratified GWAS summary statistics besides those produced through the primary analyses. Further, Supplementary Table 1 lists the genetic correlations between the male and female-specific GWAS, ranging from 0.56 for lean muscle volume index to 0.97 for muscle fat infiltration, as well as the locus yield.

## Discussion

Here, we reported results from a comprehensive, large-scale GWAS of MRI-derived measures of body composition. Joint analyses of measures of regional adipose and muscle tissue distributions revealed extensive genetic overlap and led to the identification of a large number of shared genetic risk loci across traits. We further showed genetic overlap with body anthropometrics and cardiometabolic measures as well as medical conditions linked to cardiometabolic health. Our findings illustrate how MRI-derived measures can be leveraged to improve our understanding of the biology underlying the metabolic system, emphasizing liver fat as a particularly promising measure, highlighting the integral role of steatosis and MAFLD in cardiometabolic health.

The genetic correlations of body composition measures with common medical conditions underlined that they may complement conventional measures to better understand

cardiometabolic health. Liver fat showed a stronger genetic correlation with type 2 diabetes than conventional measures. While causality needs to be established, this correlation could suggest that the amount of liver fat and its genetic determinants may play a central role in type 2 diabetes development, and at a minimum robustly positions MAFLD onto the map of relevant comorbidities of type 2 diabetes alongside cardiovascular disease, kidney disease and diabetic retinopathy. Further, we found significant positive genetic correlations between coronary artery disease and visceral and subcutaneous adipose tissue, adding genetic evidence to the well-established relation between this disease, obesity, and body fat distribution[60].

Liver fat also stood out from the other measures with regard to its genetic architecture. While all traits investigated were substantially heritable, the genetic discoverability of liver fat was much higher, with an oligogenic architecture as opposed to the polygenic architectures of the remaining traits and other complex biomedical measures[47]. This was reflected in the GWAS yield, with a few highly significant loci coupled to lipid homeostasis explaining the majority of genetic variance for this measure. These loci should be scrutinized for the biological link between liver fat and cardiometabolic conditions[61], and may potentially point to fundamental processes that become dysregulated in these diseases. Indeed, all components of metabolic syndrome correlate with liver fat content[62]. Evaluation of MAFLD risk has been recommended for any individual with metabolic syndrome and related morbidities (e.g., type 2 diabetes)[11,62], and the large effects of these liver fat-associated loci even may suggest potential as features for individual risk stratification in MAFLD[63,64]. These findings also attest to the accuracy and clinical relevance of MRI-derived measures of liver fat, and support the notion that MAFLD should be considered an integral component of obesity and metabolic syndrome and a key non-communicable disease[11].

Another key finding was that the highest number of significant loci were found for muscle fat infiltration in the anterior and posterior thighs, two measures not previously genetically studied. Fatty infiltration of skeletal muscle reduces the muscle mass and strength[65], and has been implicated in aging and frailty[66]. It has also been coupled to metabolic syndrome[67] and cardiovascular mortality[68]. Recent literature focused on liver disease and its progression have also highlighted the importance of muscle health[69]. Muscle fat infiltration has been linked to higher comorbidity within MAFLD and decreased muscle fat infiltration has been correlated with improvement in steatohepatitis[70,71]. Our findings suggest a strong genetic component to these associations, indicated by the large degree of shared genetic architecture with related diseases. Interestingly, fat accumulation in the muscle arises through specific pathways, including the intramyocellular accumulation of lipid[65], which is associated with insulin insensitivity and inflammation[72].

The genetic correlations between the MRI-derived body composition measures indicate partly overlapping biological processes with some unique genetic determinants. The correlation structure further suggests that adipose tissue distribution is genetically largely independent from muscle tissue. However, it should be noted that global correlations underestimate overlap when a mixture of genetic effects in the same and opposing directions cancels each other out[48]. Indeed, adipose and muscle tissue are known to have complex regulatory cross-talk, both releasing metabolism-regulating molecules to maintain a balanced weight-to-muscle ratio[73]. The increased yield from the multivariate GWAS analysis, nearly doubling the number of unique loci discovered, is in line with the hypothesis of strong biological interplay and shared molecular mechanisms. The multivariate GWAS allowed for identifying loci that have distributed effects across the included body composition measures. These may help to explain

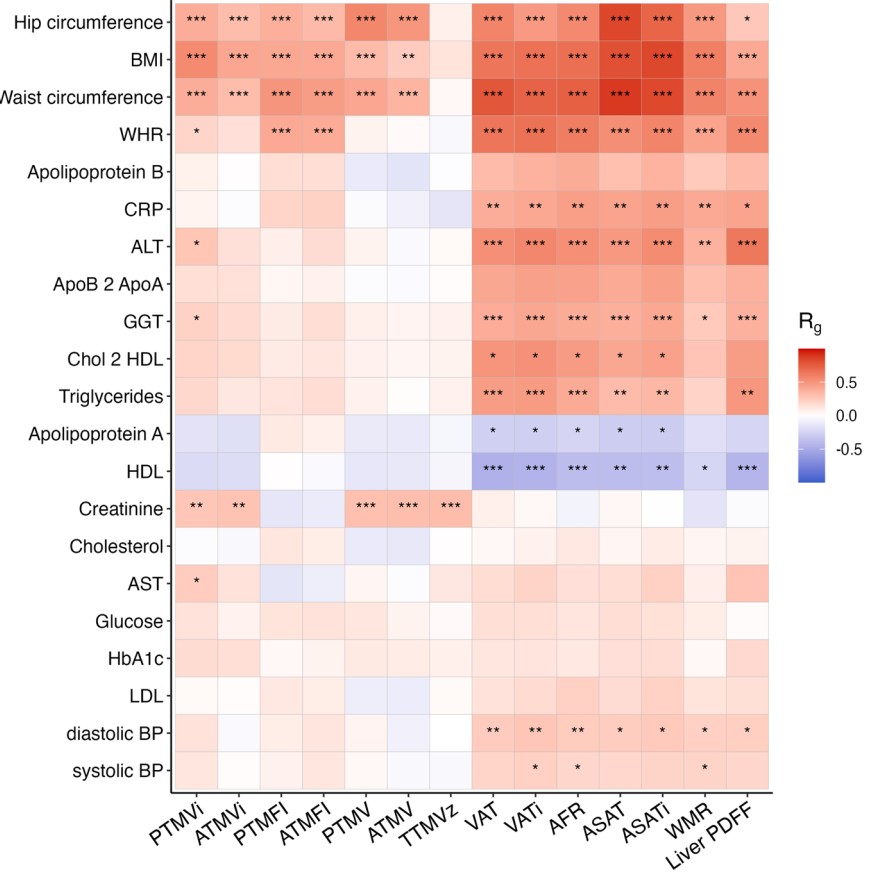

**Fig. 4 Genetic correlations of the MRI-derived body composition measures with standard anthropometrics and cardiometabolic measures.**
Abbreviations: *BMI* body mass index, *WHR* waist-hip ratio, *CRP* C-reactive protein, *ALT* alanine aminotransferase, *GGT* gamma-glutamyl transferase, *HDL* high-density lipoproteins, *AST* aspartate aminotransferase, *HbA1c* glycated hemoglobin, *LDL* low-density lipoproteins, *BP* blood pressure. ***$p = 5 \times 10^{-9}$, **$p = 5 \times 10^{-6}$, *$p = 5 \times 10^{-4}$.

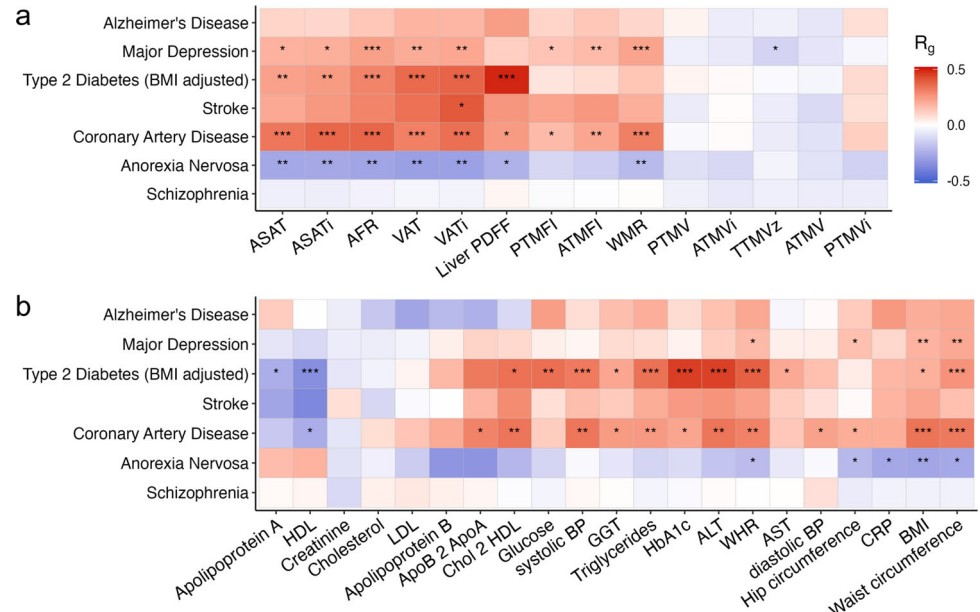

**Fig. 5 Genetic correlations with conditions linked to poor cardiometabolic health. a** Correlations for MRI-derived body composition measures on the *x*-axis. **b** These correlations for anthropometric and metabolic measure (*x*-axis) with conditions linked to poor cardiometabolic health (*y*-axis). ***$p = 5 \times 10^{-9}$, **$p = 5 \times 10^{-6}$, *$p = 5 \times 10^{-4}$.

the complexity of metabolic syndrome and the frequent comorbidity between diseases associated with body composition. Our findings that a substantial portion of the genetic determinants of these measures are related to the immune system fit with a large body of literature indicating that adipose tissue is an active metabolic and endocrine organ that secretes a host of pro- and anti-inflammatory factors, and with the characterization of obesity as a state of chronic low-grade inflammation[74]. Thus, the current genetic findings can form the basis for functional follow-up studies to determine the molecular mechanisms involved in the complex relations between lipids and the immune system.

There was high genetic overlap between the sets of MRI-derived measures of body composition and the conventional measures of body anthropometrics and cardiometabolic health, indicating that they tag similar biological processes. The body anthropometrics were correlated with both muscle and adipose tissue, indicating little specificity, in line with the long-standing recognized limitations of these global measures that they fail to distinguish between specific body types that differ widely in risk for disease[75].

Strengths of this study are the large number of whole-body MRI scans and the use of state-of-the-art, precise body composition measures, including multiple measures not previously investigated. With this, we were able to replicate loci reported earlier for MRI-derived measures of organ tissue, with different measurement protocols and different study focus[40]. We further combined the study of individual measures with a multivariate approach to genetic discovery, allowing for greater GWAS locus yield and insight into the overall architecture of these complementary indicators of body composition and associated diseases. However, further studies are needed to clarify the role of putative moderators such as sex[76], age, and ethnicity[77].

Limitations of this study include the fact that the approaches employed do not allow for causal claims beyond genetic associations. Establishing the directionality of causal effects underlying the genetic correlations between the studied measures and diseases will therefore require follow-up investigation, e.g., through Mendelian randomization. Our investigation of traits was further not comprehensive, and we lacked a sufficiently powered GWAS of MAFLD for inclusion in the analyses of genetic correlation. Further, the limited locus yield from the univariate GWAS and the low percentage of explained genetic variance for the body MRI measures, with the exception of liver fat, point towards low statistical power. The collection of larger sample sizes, as now underway through several large-scale initiatives such as the UK Biobank, and the use of more powerful statistical approaches, such as MOSTest, will be required to improve discovery.

To conclude, the high prevalence of cardiometabolic diseases, combined with substantial morbidity and mortality, indicates a strong need for new therapeutic targets. While these diseases are often comorbid, they are treated separately, with this polypharmacy bringing along increased risk of adverse drug reactions[4]. Genetic data is less subjected to reverse causation and confounding than environmental factors. Knowledge about shared and specific genetic determinants is therefore central to develop effective strategies that optimally treat the individual. Contributing to this, we showed that accurate MRI-derived measures of liver and regional adipose and muscle tissue characteristics have strong genetic components. However, our findings have also made clear that the majority of these measures are highly complex and polygenic, leading to limited yield with current sample sizes. Further, when combined, they tag similar sets of biological processes as widely available measures of anthropometrics and blood markers. This raises the question whether it is worthy to collect costly MRI scans to obtain these measures, which is hard to answer firmly with current knowledge. This study does show that the individual measures have their own unique patterns of genetic correlations and that they lead to the identification of novel loci, indicating they capture unique information, which may prove important to tease apart the influences of complex biological processes on body composition. Further, we prove that the shared influences can be leveraged to boost discovery. As such, aided by their growing availability, these measures have the potential to substantially enhance our understanding of body composition and related diseases, provide drug targets for MAFLD and related traits, and contribute to combatting a large, increasing threat to public health.

## Methods

**Participants**. We made use of data from participants of the UKB population cohort, obtained from the data repository under accession number 27412. The composition, set-up, and data gathering protocols of the UKB have been extensively described elsewhere[78]. It has received ethics approval from the National Health Service National Research Ethics Service (ref 11/NW/0382), and obtained informed consent from its participants. For the primary analyses, we selected White Europeans that had undergone the body MRI protocol, with available genetic and complete covariate data. As a final step, we excluded one of each pair of related individuals in the remaining sample, as determined through KING[79] and released by UKB, using a kinship coefficient threshold of 0.0884 ($n = 448$), leaving $N = 33\,588$, mean age 64.5 years (SD = 7.5) at the time of the MRI scan, 51.4% female. For the replication analyses, we made use of data from non-White Europeans, with otherwise same exclusion criteria, leaving $N = 5042$, mean age 63.0 years (SD = 7.7), 52.9% female. This ethnic grouping was based on self-report and confirmed by genetic principal component analysis (UKB data field 22006). For the secondary analyses of measures of cardiometabolic health, we included all White Europeans with complete genetic and covariate data. After excluding one of each pair of related individuals ($n = 34,366$), this sample consisted of 377,950 individuals with a mean age of 57.4 years (SD = 7.9), 53.7% female. The lower age of this sample reflects that we used age at baseline assessment, when these measures were taken.

**Data collection and pre-processing**. Body and liver MRI scans were collected from four scanning sites throughout the United Kingdom, all with identical scanners and protocols. They were acquired on 1.5 T Siemens MAGNETOM Aera scanners using a body dual-echo Dixon Vibe protocol and a single-slice multi-echo gradient Dixon acquisition, respectively. The UKB core neuroimaging team has published extensive information on the applied scanning protocols and procedures, which we refer to for more details[80]. We acquired the data as processed by AMRA (Linköping, Sweden; https://www.amramedical.com), subsequently released by UKB. We bridged with UKB project accession #6569 to obtain early access to this data, which was then obtained from the UKB data repositories and stored locally at the secure computing cluster of the University of Oslo.

The methods used to generate the MRI-derived measurements has been described and evaluated in more detail elsewhere[1–6]. Briefly, the process for fat and muscle compartments includes the following steps: (1) calibration of fat images using fat-referenced MRI, (2) registration of atlases with ground truth labels for fat and muscle compartments to the acquired MRI dataset to produce automatic segmentation, (3) quality control by two independent trained operators including the possibility to adjust and approve the final segmentation, and (4) quantification of fat volumes, muscle volumes and muscle fat infiltration within the segmented regions. For liver proton density fat fraction (PDFF), nine regions of interest (ROI) were manually placed, evenly distributed in the liver volume, while avoiding major vessels and bile ducts.

Muscle volumes were calculated as fat-tissue free muscle volumes. Muscle fat infiltration was calculated as the average T2*-corrected fat fraction and converted to PDFF[4,7]. Liver PDFF was calculated depending on the protocols implemented by UK Biobank. The liver protocol was initially based on a single-slice symmetric 10-point acquisition with IDEAL reconstruction, but after 10.000 scans it was replaced by the whole-body dual echo Dixon images with an additional T2* and proton density correction, where the T2* values were estimated from a separate single-slice asymmetric 6-point acquisition. The latter method has been described and validated against IDEAL-based liver PDFF previsouly[4]. 1.338 scans were acquired with both liver protocols to assess the switch, and a good agreement was found between the protocols—the mean difference in liver PDFF was 0.30% with a standard deviation of the differences of 0.80%.

Test-retest reliability of the MRI-derived measures included in this study is high, with a nearly perfect intraclass correlation coefficient, and the automated processing performs better than manual segmentation of muscle and fatty tissue[8,9]. A recent study, investigating both regional fat and muscle volumes as well as muscle fat infiltration and liver fat fraction, showed high repeatability and reproducibility on five different 1.5 T and 3 T scanners from three different vendors[4].

**Measurement protocols and definitions**. We extracted a selection of body composition measures (Table 1; see also UKB online documentation (http://biobank.ctsu.ox.ac.uk/showcase/)). Specifically, we extracted the following measures of adipose tissue: VAT, defined as the adipose tissue within the abdominal cavity, and abdominal subcutaneous adipose tissue (ASAT), defined as the adipose tissue between the top of the femoral head and the top of T9. We also extracted measures of muscle fat infiltration (MFI) derived from the anterior and posterior thighs (anterior thighs including quadriceps femoris, sartorius, and tensor fascia latae, and posterior thighs including gluteus muscles, iliacus, adductor muscles, and hamstring muscles), averaged over both legs, and liver PDFF. As measures of muscle tissue, we included anterior and posterior thigh fat-free muscle volume (ATMV and PTMV). We extracted two ratios from the UKB repository, namely weight-to-muscle ratio (WMR), defined as weight/TTMV, and abdominal fat ratio (AFR), which is (VAT + ASAT)/(VAT + ASAT + TTMV). For VAT, ASAT, ATMV, and PTMV, we computed index measures by dividing these measures by the squared standing height in meters (e.g., ASATi is ASAT/height$^2$). This is done since weight, adipose tissue, and lean tissue compartments scale to approximate height squared. In addition, a sex-, height-, and weight invariant normalized $z$-score for TTMV (TTMVz) was calculated. Including adjustment for sex, height, and weight has been shown to strengthen the association between muscle volume and hospitalization/function, and TTMVz has previously been associated with poor function, hospitalization, and all-cause mortality in general population, as well as poor function and metabolic comorbidity in MAFLD[10–12].

We subsequently regressed out age, sex, scanner site, genotyping array and the first twenty genetic principal components from each measure. Following this, we applied rank-based inverse normal transformation[81] to the residuals of each measure, leading to normally distributed measures as input for the GWAS.

For the secondary analyses, comparing the set of MRI-derived measures of body composition to measures of cardiometabolic health, we included 21 measures available in the UKB as listed in Table 2.

**GWAS procedure**. We made use of the UKB v3 imputed data, based on two highly similar genotyping arrays (UK BiLEVE and UKB Axiom), which has undergone extensive quality control procedures as described by the UKB genetics team[82]. After converting the BGEN format to PLINK binary format, we additionally carried out standard quality check procedures, including filtering out individuals with more than 10% missingness, SNPs with more than 5% missingness, and SNPs failing the Hardy–Weinberg equilibrium test at $p = 1 \times 10^{-9}$. We further set a minor allele frequency threshold of 0.005, leaving 9,061,022 SNPs.

**Statistics and reproducibility**. We carried out GWAS through the freely available MOSTest software (https://github.com/precimed/mostest), with the approaches employed being identical for both the primary analyses of the body MRI data and the secondary analyses of cardiometabolic health metrics. The procedure has been extensively validated[49]. GWAS on each of the pre-residualized and normalized measures were carried out using the standard additive model of linear association between genotype vector, $g_j$, and phenotype vector, $y$. Independent significant SNPs and genomic loci were identified in accordance with the PGC locus definition, as also used in FUMA SNP2GENE[83]. First, we selected a subset of SNPs that passed genome-wide significance threshold $5 \times 10^{-8}$, and used PLINK to perform a clumping procedure at linkage disequilibrium (LD) $r^2 = 0.6$ to identify the list of independent significant SNPs. Second, we clumped the list of independent significant SNPs at LD $r^2 = 0.1$ threshold to identify lead SNPs. Third, we queried the reference panel for all candidate SNPs in LD $r^2$ of 0.1 or higher with any lead SNPs. Further, for each lead SNP, its corresponding genomic loci were defined as a contiguous region of the lead SNPs' chromosome, containing all candidate SNPs in $r^2 = 0.1$ or higher LD with the lead SNP. Finally, adjacent genomic loci were merged if separated by less than 250 KB. Allele LD correlations were computed from EUR population of the 1000 genomes Phase 3 data. We made use of the Functional Mapping and Annotation of GWAS (FUMA) online platform (https://fuma.ctglab.nl/) to map significant SNPs from the MOSTest analyses to genes. For this, we combined the default positional mapping with eQTL and 3D chromatin interaction mapping, including all available tissue types.

**MiXeR analysis**. We applied a causal mixture model[45,46] to estimate the percentage of variance explained by genome-wide significant SNPs as a function of sample size. For each SNP, $i$, MiXeR models its additive genetic effect of allele substitution, $\beta_i$, as a point-normal mixture, $\beta_i = (1 - \pi_1)N(0, 0) + \pi_1 N(0, \sigma_\beta^2)$, where $\pi_1$ represents the proportion of non-null SNPs ('polygenicity') and $\sigma_\beta^2$ represents the variance of effect sizes of non-null SNPs ('discoverability'). Then, for each SNP, $j$, MiXeR incorporates LD information and allele frequencies for 9,997,231 SNPs extracted from 1000 Genomes Phase3 data to estimate the expected probability distribution of the signed test statistic, $z_j = \delta_j + \epsilon_j = N \sum_i \sqrt{H_i r_{ij}} \beta_i + \epsilon_j$, where $N$ is the sample size, $H_i$ indicates heterozygosity of i-th SNP, $r_{ij}$ indicates an allelic correlation between i-th and j-th SNPs, and $\epsilon_j \sim N(0, \sigma_0^2)$ is the residual variance. Further, the three parameters, $\pi_1, \sigma_\beta^2, \sigma_0^2$, are fitted by direct maximization of the likelihood function. Fitting the univariate MiXeR model does not depend on the sign of $z_j$, allowing us to calculate

$|Z_j|$ from MOSTest $p$-values. Finally, given the estimated parameters of the model, the power curve $S(N)$ is then calculated from the posterior distribution $p(\delta_j | z_j, N)$.

**LD score regression**. For estimates of SNP-based heritability ($h_2$), we applied LD score regression (LDSR)[84] to the univariate GWAS summary statistics. For this, each set of summary statistics underwent additional filtering, including the removal of all SNPs in the extended major histocompatibility complex region (chr6:25–35 Mb). We further used these munged summary statistics to perform cross-trait LDSR to estimate genetic correlations between the measures[85].

**Gene-set analyses**. We carried out gene-based analyses using MAGMA v1.08 with default settings, which entails the application of a SNP-wide mean model and the use of the 1000 Genomes Phase 3 EUR reference panel. Gene-set analyses were done in a similar manner, restricting the sets under investigation to those that are part of the Gene Ontology biological processes subset ($n = 7522$), as listed in the Molecular Signatures Database (MsigdB; c5.bp.v7.1).

**Reporting summary**. Further information on research design is available in the Nature Portfolio Reporting Summary linked to this article.

## Data availability

The data incorporated in this work were gathered from public resources, with UK Biobank data repository under accession number 27412. GWAS summary statistics are uploaded to the GWAS catalog (https://www.ebi.ac.uk/gwas/). Source data for Supplementary Fig. 4 is provided in Supplementary Data 7. Correspondence and requests for materials should be addressed to d.v.d.meer@medisin.uio.no.

## Code availability

The code used in this project is freely available via https://github.com/precimed/mostest (GPLv3 license) and https://github.com/norment/open-science.

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

## Acknowledgements

This work was partly performed on the TSD (Tjeneste for Sensitive Data) facilities, owned by the University of Oslo, operated and developed by the TSD service group at the University of Oslo, IT-Department (USIT). (tsd-drift@usit.uio.no). Computations were also performed on resources provided by UNINETT Sigma2 - the National Infrastructure for High-Performance Computing and Data Storage in Norway. The authors were funded by the Research Council of Norway (276082, 213837, 223273, 204966/F20, 229129, 249795/F20, 225989, 248778, 249795, 298646, 300767), the South-Eastern Norway Regional Health Authority (2013-123, 2014-097, 2015-073, 2016-064, 2017-004, 2017-112, 2019-101, 2020-060 (IES), 2022-080), Stiftelsen Kristian Gerhard Jebsen (SKGJ-MED-021), The European Research Council (ERC) under the European Union's Horizon 2020 research and innovation program (ERC Starting Grant, Grant agreement No. 802998), ERA-Net Cofund through the ERA PerMed project 'IMPLEMENT', and National Institutes of Health (R01MH100351, R01GM104400). This project has received funding from the European Union's Horizon 2020 Research and Innovation Program under Grant agreement No 847776 (CoMorMent).

## Author contributions

D.v.d.M., T.G., I.E.S., and O.A.A. conceived the study; D.v.d.M., T.G., and T.K. pre-processed the data. D.v.d.M. performed all analyses, with conceptual input from O.A.A.; D.v.d.M., T.G., I.E.S., A.A.S., G.H., Z.R., A-M.G.L, O.F., O.D.L., J.L., R.S., D.B., L.T.W., S.H., A.M.D., T.H.K., T.K., and O.A.A. contributed to interpretation of results; D.v.d.M. drafted the paper and T.G., I.E.S., A.A.S., G.H., Z.R., A-M.G.L, O.F., O.D.L., J.L., R.S., D.B., L.T.W., S.H., A.M.D., T.H.K., T.K., and O.A.A. contributed to and approved the final paper.

## Competing interests

The authors declare the following competing interests: O.A.A. has received speaker's honorarium from Lundbeck and is a consultant to HealthLytix. J.L. and O.D.L. are employed by and stockholders in AMRA Medical, Inc., and R.S. was previously employed by AMRA medical. T.H.K. received consultancy fees from Intercept and Engitix and speaker fees from Novartis, Gilead and AlfaSigma. A.M.D. is a Founder of and holds equity in CorTechs Labs, Inc., and serves on its Scientific Advisory Board. He is a member of the Scientific Advisory Board of Human Longevity, Inc. and receives funding through research agreements with General Electric Healthcare and Medtronic, Inc. The terms of these arrangements have been reviewed and approved by UCSD in accordance with its conflict of interest policies. The remaining authors declare no competing interests.

## Additional information

[1]Norwegian Centre for Mental Disorders Research (NORMENT), Division of Mental Health and Addiction, Oslo University Hospital & Institute of Clinical Medicine, University of Oslo, Oslo, Norway. [2]School of Mental Health and Neuroscience, Faculty of Health, Medicine and Life Sciences, Maastricht University, Maastricht, The Netherlands. [3]Department of Medical Genetics, Oslo University Hospital, Oslo, Norway. [4]K.G. Jebsen Centre for Neurodevelopmental Disorders, University of Oslo, Oslo, Norway. [5]Psychosis Studies, Institute of Psychiatry, Psychology and Neurosciences, King's College London, London, UK. [6]LREN, Centre for Research in Neurosciences, Dept. of Clinical Neurosciences, Lausanne University Hospital (CHUV) and University of Lausanne, Lausanne, Switzerland. [7]Dept. of Psychiatry, University of Oxford, Oxford, UK. [8]Centre for Bioinformatics, Department of Informatics, University of Oslo, Oslo, Norway. [9]AMRA Medical, Linköping, Sweden. [10]Division of Diagnostics and Specialist Medicine, Department of Health, Medicine and Caring Sciences, Linköping University, Linköping, Sweden. [11]Center for Medical Image Science and Visualization (CMIV), Linköping University, Linköping, Sweden. [12]Department of Psychology, University of Oslo, Oslo, Norway. [13]Department of Psychiatric Research, Diakonhjemmet Hospital, Oslo, Norway. [14]Department of Cardiology, Oslo University Hospital Ullevål, and University of Oslo, Oslo, Norway. [15]Center for Multimodal Imaging and Genetics, University of California at San Diego, La Jolla, CA 92037, USA. [16]Department of Transplantation Medicine, Division of Surgery, Inflammatory Diseases and Transplantation, Oslo University Hospital Rikshospitalet, Oslo, Norway. [17]Research Institute for Internal Medicine, Division of Surgery, Inflammatory Diseases and Transplantation, Oslo University Hospital Rikshospitalet and University of Oslo, Oslo, Norway. [18]Department of Psychiatry and Psychotherapy, University of Tübingen, Tübingen, Germany. ✉email: d.v.d.meer@medisin.uio.no

