## [Peer Review File · Communications Biology]

Reviewers' comments:

Reviewer #1 (Remarks to the Author):

In this manuscript, van der Meer et al performed univariate and multivariate GWAS using MRI derived measures of body composition including different measures of fat deposition and muscle volume. By estimating the genetic correlation between these MRI derived measures of body composition and metabolic biomarkers and metabolic disease outcomes, they showed liver fat plays a key role in cardiometabolic health.

In the results:

It says 'The Supplementary Information (SI) contains Manhattan plots'. I could not find these plots or any other supplementary figures in the supplementary files.

Please provide confidence interval when you provide information on mean.

The univariate and multivariate GWAS signals are not presented and discussed.

In the method:

How did you define white Europeans and non-white Europeans?

The method they used to quantify different fat and muscle measures from MRI scan is not clear. How reliable are these measures? How did they train their algorithm to generate these measures?

It is not clear why some measures are corrected for BMI (e.g. TTMV) and some measures are not.

The GWAS method section does not address how they have excluded related participants and if not, how they corrected for relatedness.

Genotyping arrays should be used as a covariate in the GWAS model.

There is no information on how the GWAS of metabolic biomarkers and anthropometric traits were performed.

In the discussion:

Some statements are not correct and need to be modified. For example a genetic correlation between liver fat and type 2 diabetes does not mean that liver fat has a causal role in type 2 diabetes development. This conclusion cannot be driven from the results of this study.

Reviewer #2 (Remarks to the Author):

The manuscript by van der Meer et al reports GWAS on 14 MRI-derived measures of adipose and muscle tissue distribution in a large population of participants from the UKBB (~34K). The authors report an important data processing of imaging to estimate body composition of participants including various locations of fat and muscle in the abdomen and thighs. They also estimated fat infiltration in muscles and liver and extracted weight to muscle ratio and abdominal fat ratio from UKBB repository. The authors complemented their work by conducting multivariate analyses including these measures

combined with cardiometabolic traits of the same cohort, which were available in much larger samples (>10 times the MRI-derived samples). The authors report a large number of loci from this multivariate analysis, which is expected from such large samples. Overall, this is a well conducted study, especially for phenotype definition based on imaging, and this is where the most original data is reported. The genetic methods are standard GWAS methods, sound and well applied. The manuscript is overall well-written, although the introduction could be shortened as well as the discussion. However, the study lacks a clear conclusion to be driven from their results about the usefulness of using these very complex estimations of fat liver content for instance. After reading the article, one continues to wonder if it worth it to perform an MRI if we can have genetic metabolic risk estimated from BMI, liver enzymes...etc.

1. The authors are invited to explicitly indicate if there are any novel loci identifies using these extremely complex phenotypes, compared to classical biochemical measures of liver function using comparable sample sizes. Overall, the number of loci is extremely low, supporting that these MRI-derived traits are just as imprecise as other anthropometric traits. Could the authors comment more explicitly on this lack of power of to identify novel loci?
2. What about biochemical enzymes measured in blood? Do the authors find most or at least some of the loci identified by GWAS for ALT, GGT and AST?
3. The authors state that heritability estimated of MRI-derived measures were higher on average compared to anthropometrics and biomarkers. Would be possible to provide specific estimates for each trait cited? Indicating average heritability is not precise enough to get the full picture about how heritable are these traits, especially compared to biochemically measured levels of biomarkers of liver function.
4. What is the rationale of performing genetic correlations estimations with Major depression, Anorexia nervosa and Schizophrenia? Why, in their opinion, the shared genetics is more precisely estimated with MRI-measures compared to anthropometrics? One would expect that anthropometric measures of obesity cover also behavioral causes of high weight, including brain-controlled food intake, not better assessment of fat content in muscle or liver. Please comment on this.
5. Do the authors see any differences between estimations of fat content between men and women? If they have access to this data, stratified analyses by sex, and /or interaction with sex would be informative given how fat content is different according to sex.
6. As they mention strength of the study, the authors are invited to provide a limitation section of their study, with mention of the overall lack of power to identify additional loci different from those covered by classical cardiometabolic traits.

Reviewer #3 (Remarks to the Author):

The manuscript describes briefly the genetic architecture of 31 traits extracted from large-scale MRI conducted in the UK Biobank. There are now several large studies looking at some of these and related traits (e.g. <https://www.nature.com/articles/s41591-019-0563-7> - there are many others) and investigating the contribution of these traits to cardiometabolic disease using genetic epidemiology. While several of the traits, especially those related to muscle composition, have not to my knowledge been studied at this scale before, I find the study overstates its novelty a little in this regard. A major finding is the genetic correlation between liver fat and type 2 diabetes pointing to a possible etiological role; this is also not a new hypothesis (e.g. <https://diabetesjournals.org/care/article-abstract/45/2/460/139118/Estimating-the-Effect-of-Liver-and-Pancreas-Volume>) The novel aspects in my opinion are: (1) definition of several traits and indices related to body composition; (2) integration of many adiposity and body composition traits into a single measure and conducting a multivariate analysis using software previously developed by the authors; (3) semi-systematic genetic correlation with metabolic risk factors and several disease outcomes, establishing and exploring the relationships between these traits.

1. Given the focus on MAFLD (is this the same as NAFLD?) in the introduction and discussion, it is

somewhat surprising that this was not included in the genetic correlation analysis. How were the diseases of interest chosen?

2. The discussion of the identified genes is very minimal. There have been several previously described studies in this cohort looking at related traits. How many of the loci identified in the univariate analysis are novel?

3. The methods do not adequately describe the generation of the MRI-derived measurements, and the reference does not give a complete picture either. What "manual adjustments" were performed? Were they performed for all 35,000 participants? The derivation of PDFF is also not adequately described and no reference is given here. A description of quality control processes (if any) is also missing.

4. On the genetics side, several sections are missing detailed methods: heritability estimation, genetic correlation methods, annotation of genes, etc. Also QC/sample inclusion criteria are not fully described - for example, were close relatives removed?

5. A major finding of the study concerns the heritability and genetic architecture of the adiposity and muscle traits. The UK Biobank is known to oversample related individuals (<https://www.nature.com/articles/s41586-018-0579-z>) relative to their frequency in the general population. The method used does not account for this relatedness and is thus vulnerable to an inflation of type-1 error rate (presumably this would also affect the downstream analysis). I suggest that the authors check for residual confounding using one of the available standard methods (e.g. genomic control, LDSC regression intercept

(<https://www.ncbi.nlm.nih.gov/pmc/articles/PMC6758917/>), or a QQ plot). In addition, correcting for covariates before rank normalization can re-introduce confounding (<https://www.nature.com/articles/s41431-018-0159-6>)

6. I don't see a data availability statement in the manuscript, although the box is checked. I encourage the authors to make their summary statistics available via GWAS catalog <https://www.ebi.ac.uk/gwas/>

We are grateful for the chance to revise our manuscript and we wish to thank the reviewers for their insightful feedback. Below, we provide a point-by-point response to their comments and indicate how we have updated the manuscript in accordance with these points. Overall, we have substantially improved the analytical approach, including correction for relatedness. We have also enhanced the description of methodological details as requested by the reviewers and increased the amount of information provided on the results from the GWAS. We believe these adjustments have significantly improved the study and the manuscript.

Reviewer #1 (Remarks to the Author):

In this manuscript, van der Meer et al performed univariate and multivariate GWAS using MRI derived measures of body composition including different measures of fat deposition and muscle volume. By estimating the genetic correlation between these MRI derived measures of body composition and metabolic biomarkers and metabolic disease outcomes, they showed liver fat plays a key role in cardiometabolic health.

In the results:

It says 'The Supplementary Information (SI) contains Manhattan plots'. I could not find these plots or any other supplementary figures in the supplementary files.

This is unfortunate, thank you for pointing out this technical error. We will work with the editorial staff to ensure the supplementary document containing all figures and accompanying descriptions will be clearly marked and available to the reviewers.

Please provide confidence interval when you provide information on mean.

We agree that including confidence intervals is good practice and apologize for omitting them in the original manuscript. We now include them in the Results section as follows:

“The heritability of the MRI-derived measures (mean $h^2=.25$, 95% CI [.22, .28]) was significantly higher than the body anthropometrics and other biomarkers (mean $h^2=.13$, 95% CI [.10, .15]), $p=1.8 \times 10^{-7}$.”

The univariate and multivariate GWAS signals are not presented and discussed.

Thank you for this comment. Our choice of what analyses to describe was driven by the fact that this study involves the investigation of multiple sets of measures, with the goal to provide an overview of their overall genetic architectures. We thereby present the number of loci discovered per measure, the statistical power associated with these measures, and the genetic overlap between them and other traits. The corresponding Manhattan plots as well as lists of loci and enriched pathways are in supplementary material, as unfortunately there is no room to cover this in detail in the main manuscript. We have now further included additional information on the heritability estimates per measure, plus standard errors, for all individual measures in the Supplementary Material, Figure 4.

In the method:

How did you define white Europeans and non-white Europeans?

Apologies, we should have described how we defined White Europeans. This has now been rectified by including the following text in the Methods section, under the ‘Participants’ header:

“This ethnic grouping was based on self-report and confirmed by genetic principal component analysis (UKB data field 22006).”

The method they used to quantify different fat and muscle measures from MRI scan is not clear. How reliable are these measures? How did they train their algorithm to generate these measures?

Given that the focus of the current study is on the genetic architecture of these measures, we chose for the original manuscript to reference previous publications that describe the methodology used to obtain these measures and their validity. However, we agree that this is important information, and that it warrants a succinct description to enable readers to judge the value of the findings. Therefore, we now include the following text in the revised Methods section:

“The methods used to generate the MRI-derived measurements has been described and evaluated in more detail elsewhere.¹⁻⁶ Briefly, the process for fat and muscle compartments includes the following steps: (1) calibration of fat images using fat-referenced MRI, (2) registration of atlases with ground truth labels for fat and muscle compartments to the acquired MRI dataset to produce automatic segmentation, (3) quality control by two independent trained operators including the possibility to adjust and approve the final segmentation, and (4) quantification of fat volumes, muscle volumes and muscle fat infiltration within the segmented regions. For liver proton density fat fraction (PDFF), nine regions of interest (ROI) were manually placed, evenly distributed in the liver volume, while avoiding major vessels and bile ducts.

Muscle volumes were calculated as fat-tissue free muscle volumes. Muscle fat infiltration was calculated as the average T2*-corrected fat fraction and converted to proton density fat fraction.^{4,7} Liver PDFF was calculated depending on the protocols implemented by UK Biobank. The liver protocol was initially based on a single-slice symmetric 10-point acquisition with IDEAL reconstruction, but after 10,000 scans it was replaced by the whole-body dual echo Dixon images with an additional T2* and proton density correction, where the T2* values were estimated from a separate single-slice asymmetric 6-point acquisition. The latter method has been described and validated against IDEAL-based liver PDFF previously.⁴ 1,338 scans were acquired with both liver protocols to assess the switch, and a good agreement was found between the protocols - the mean difference in liver PDFF was 0.30% with a standard deviation of the differences of 0.80%.

Test-retest reliability of the MRI-derived measures included in this study is high, with a nearly perfect intraclass correlation coefficient, and the automated processing performs better than manual segmentation of muscle and fatty tissue.^{8,9} A recent study, investigating both regional fat and muscle volumes as well as muscle fat infiltration and liver fat fraction, showed high repeatability and reproducibility on five different 1.5T and 3T scanners from three different vendors.⁴”

It is not clear why some measures are corrected for BMI (e.g. TTMV) and some measures are not.

We apologize for the lack of explanation why some measures are corrected for BMI and others are not. It has been shown that correcting TTMV for BMI through calculation of a normalized z-score according to Linge et al 2019 strengthens the association between muscle volume and function/hospitalization.¹⁰ This measurement has also, in more recent work, been associated with poor function and metabolic comorbidity within NAFLD, as well as all-cause mortality within general population.¹¹ We have now restructured the paragraph in the Methods section about ‘Measurement protocols and definitions’ slightly and included the following text to clarify this:

“In addition, a sex-, height-, and weight invariant normalized z-score for TTMV (TTMVz) was calculated. Including adjustment for sex, height, and weight has been shown to strengthen the association between muscle volume and hospitalization/function, and TTMVz has previously been associated with poor function, hospitalization, and all-cause mortality in general population, as well as poor function and metabolic comorbidity in MAFLD.¹⁰⁻¹²”

Also worth noting based on our results is that in contrast to the other lean tissue measures, TTMVz did not show any genetic correlation with body anthropometrics, suggesting that this measure captures the genetics of lean tissue while factoring out the body size and shape.

The GWAS method section does not address how they have excluded related participants and if not, how they corrected for relatedness.

Thank you for pointing this out. We did not correct for relatedness in the initial analyses. We have now improved the analysis according to Reviewer's comment. In the revised manuscript we now take the participant relatedness information (calculated via the KING software) as released by the UK Biobank into account and removed one of each pair of related individuals. This is described as follows in the revised 'Methods' section:

"As a final step, we excluded one of each pair of related individuals in the remaining sample, as determined through KING¹³ and released by UKB, using a kinship coefficient threshold of 0.0884 (n=448), leaving N=33 588..."

"For the secondary analyses of measures of cardiometabolic health, we included all White Europeans with complete genetic and covariate data. After excluding one of each pair of related individuals (n=34,366), this sample consisted of 377,950 individuals..."

Genotyping arrays should be used as a covariate in the GWAS model.

We have rerun the analyses with genotyping array (BiLEVE versus Axiom) included as covariate, as described in the revised manuscript:

"We pre-residualized all measures for age, sex, test center, **genotyping array**, and the first twenty genetic principal components to control for population stratification."

"We made use of the UKB v3 imputed data, **collected through two genotyping arrays (UK BiLEVE and UKB Axiom)**,..."

There is no information on how the GWAS of metabolic biomarkers and anthropometric traits were performed.

These GWAS were carried out with identical procedures as the primary GWAS of body MRI data. To clarify this, we have edited the relevant sentence in the Methods section to explicitly state this:

"We carried out GWAS through the freely available MOSTest software (<https://github.com/precimed/mostest>), with identical approaches for both the primary analyses of the body MRI data and the secondary analyses of cardiometabolic health metrics."

In the discussion:

Some statements are not correct and need to be modified. For example a genetic correlation between liver fat and type 2 diabetes does not mean that liver fat has a causal role in type 2 diabetes development. This conclusion cannot be driven from the results of this study.

We apologize that our phrasing implied such a causal link. We have now rewritten the discussion section to minimize this implication, and have added a limitations section that calls for studies allowing for causal inference:

"While causality needs to be established, this correlation could suggest that the amount of liver fat and its genetic determinants may play a central role in type 2 diabetes development"

"Limitations of this study include that the analyses employed do not allow for causal claims beyond genetic associations. Establishing the directionality of causal effects underlying the genetic correlations

between the studied measures and diseases will therefore require follow-up investigation, e.g., through Mendelian randomization.”

Reviewer #2 (Remarks to the Author):

The manuscript by van der Meer et al reports GWAS on 14 MRI-derived measures of adipose and muscle tissue distribution in a large population of participants from The UKBB (~34K). The authors report an important data processing of imaging to estimate body composition of participants including various locations of fat and muscle in the abdomen and thighs. They also estimated fat infiltration in muscles and liver and extracted weight to muscle ratio and abdominal fat ratio from UKBB repository. The authors complemented their work by conducting multivariate analyses including these measures combined with cardiometabolic traits of the same cohort, which were available in much larger samples (>10 times the MRI-derived samples). The authors report a large number of loci from this multivariate analysis, which is expected from such large samples. Overall, this is a well conducted study, especially for phenotype definition based on imaging, and this is where the most original data is reported. The genetic methods are standard GWAS methods, sound and well applied. The manuscript is overall well-written, although the introduction could be shortened as well as the discussion. However, the study lacks a clear conclusion to be driven from their results about the usefulness of using these very complex estimations of fat liver content for instance. After reading the article, one continues to wonder if it worth it to perform an MRI if we can have genetic metabolic risk estimated from BMI, liver enzymes...etc.

Thank you for this kind summary. We agree that the paper would benefit from a more explicit conclusion on the value of the studied metrics, and we have addressed this in the discussion, as listed here and further explained in response to the specific, numbered comments below.

““However, our findings have also made clear that the majority of these measures are highly complex and polygenic, leading to limited yield with current sample sizes. Further, when combined, they tag similar sets of biological processes as widely available measures of anthropometrics and blood markers. This raises the question whether it is worthy to collect costly body MRI scans to obtain these measures, which is hard to answer firmly with current knowledge. This study does show that the individual measures have their own unique patterns of genetic correlations and that they lead to the identification of novel loci, indicating that they capture unique information, which may prove important to tease apart the influences of complex biological processes on body composition.”

1. The authors are invited to explicitly indicate if there are any novel loci identifies using these extremely complex phenotypes, compared to classical biochemical measures of liver function using comparable sample sizes. Overall, the number of loci is extremely low, supporting that these MRI-derived traits are just as imprecise as other anthropometric traits. Could the authors comment more explicitly on this lack of power of to identify novel loci?

We agree that an overview of the novelty of the loci discovered for the body MRI measures is valuable. We therefore ran additional analyses whereby we limited the secondary measures to individuals that had undergone the body MRI scans, and checked the p-values of these loci in the output. In the revision we added the Supplementary Figure below to the supplement. This makes it clear that most of the discovered loci from our primary analyses were not whole-genome significant for the other included cardiometabolic health measures at comparable sample sizes. We now describe this in the revised Results section as follows.

“To establish whether the loci discovered through the univariate GWAS of the body MRI measures are novel compared to related measures of cardiometabolism, we additionally ran univariate GWAS on 21 secondary measures of anthropometric and cardiometabolic factors (e.g., BMI, triglycerides, cholesterol, blood pressure; see Table 2). This revealed that, when restricted to the same sample of individuals with available body MRI data (N=33,588), the large majority of loci were indeed novel. Supplementary Figure 3 summarizes this and shows for each discovered variant whether it was whole-genome significant for each of the primary and secondary measures.”

Supplementary Figure 3. Novelty of loci discovered through the univariate GWAS of body MRI measures. The rs-codes on the y-axis represent the lead SNPs of the discovered loci, and on the x-axis are all the studied measures; on the left of the vertical black line are the body MRI measures and on the right are the secondary measures of cardiometabolism. Red cells indicate that the SNP is associated with the measure beyond the genome-wide significance threshold.

We acknowledge that the number of loci discovered for most of the body MRI measures is in the same range as those identified through anthropometric measures at comparable sample sizes. Rather than increased locus discovery, our message is that the body MRI measures are likely to provide complementary information to anthropometric measures, as suggested by differing patterns of genetic correlations. We also acknowledge the current limited statistical power for the body MRI measures, and have included the statement that these are complex measures that require larger sample sizes than currently available to uncover a substantial proportion of their genetic determinants as follows.

“Further, the limited locus yield from the univariate GWAS and the low percentage of explained genetic variance for the body MRI measures, with the exception of liver fat, point towards low statistical power. The collection of larger sample sizes, as now underway through several large-scale initiatives such as the UK Biobank, and the use of more powerful statistical approaches will be required to improve discovery.”

2. What about biochemical enzymes measured in blood? Do the authors find most or at least some of the loci identified by GWAS for ALT, GGT and AST?

Please see the answer for comment 1) above, and its accompanying Supplementary Figure 3. The majority of loci discovered through the GWAS on our primary measures are not whole-genome significantly associated with the biochemical measures at comparable sample sizes. As can be deduced from the Supplementary Figure 3, of the 50 unique loci discovered through the univariate analyses of the body MRI measures, we found two loci that were also significant for ALT, one for GGT, and one for AST, all overlapping with those found for liver fat. As described above, information about the novelty of the loci is now added to the revised manuscript.

3. The authors state that heritability estimated of MRI-derived measures were higher on average compared to anthropometrics and biomarkers. Would be possible to provide specific estimates for each trait cited? Indicating average heritability is not precise enough to get the full picture about how heritable are these traits, especially compared to biochemically measured levels of biomarkers of liver function.

Thank you for this suggestion. We agree that an overview of the heritability estimates for each trait is informative. We now provide a bar plot (Supplementary Figure 4) that provides these estimates to clarify the difference in heritability between the categories. Please see the newly added Supplementary Figure 4 below.

Supplementary Figure 4. Bar plot summarizing the SNP-based heritability (y-axis) for each of the univariate measures analyzed (x-axis), as calculated through LD score regression. The fill of the bars indicates the measurement category, as indicated in the legend. The error bars reflect standard error.

4. What is the rationale of performing genetic correlations estimations with Major depression, Anorexia nervosa and Schizophrenia? Why, in their opinion, the shared genetics is more precisely estimated with MRI-measures compared to anthropometrics? One would expect that anthropometric measures of obesity cover also behavioral causes of high weight, including brain-controlled food intake, not better assessment of fat content in muscle or liver. Please comment on this.

These brain disorders were included as they are among the most costly and debilitating conditions in the world, combined with the fact that our previous work has shown a strong phenotypic relation between brain morphology, related disorders, and body composition. Research such as conducted here, illustrating overlap between body and brain, may promote research that enables the development of

effective intervention, by targeting shared pathways. This reasoning is now made more explicit as follows:

“As shown by our work, body composition is also strongly associated with brain structure and brain disorders, which are among the most costly and debilitating medical conditions in the world.¹⁴⁻¹⁶”

“We further sought to identify the extent of genetic overlap between these measures and common medical conditions, as such information promotes research into shared molecular pathways and therefore a better understanding of the underlying biology.”

Genetic investigation of anthropometric measures may indeed cover behavioral influences as well as a range of different influences on overall body composition, i.e., these measures may represent a rather mixed bag of biological processes. MRI-derived measures of local adiposity or muscle volume could have a narrower set of influences than anthropometric measures and therefore provide complementary targets for investigation. For clarification, we have now rewritten the reasoning in the revised manuscript as follows:

“Body anthropometrics such as waist circumference and body mass index (BMI) lack a direct connection to pathophysiology^{17,18} Measures of regional adipose tissue, most accurately and comprehensively identified through MRI,^{19,20} may offer sensitive proxies of cardiometabolic health and therefore complement these common measures.²¹ This is further suggested by research indicating they have independent associations with cardiometabolic diseases and improve risk prediction beyond body anthropometrics.²²⁻²⁴”

5. Do the authors see any differences between estimations of fat content between men and women? If they have access to this data, stratified analyses by sex, and /or interaction with sex would be informative given how fat content is different according to sex.

Indeed, there is a large literature on sex differences in body composition, making it highly likely that there are substantial differences between men and women with regard to the biological determinants of these measures. While this is highly interesting, we chose to not make this a central topic, for sake of focus of the manuscript. Nonetheless, given that we have access to the data, we have decided to generate the GWAS summary statistics, stratified by sex, and we will make these available upon publication of the study through the GWAS Catalog, per the Data Availability statement. Further, we have included an overview of the genetic correlations between the two sets of sex-specific summary statistics, and the number of loci discovered for men versus for women. Please see below for this work, now added to the revised Results section.

“Sex-specific analyses

Given that the body composition of men and women differs substantially, we provide sex-stratified GWAS summary statistics besides those produced through the primary analyses. Further, Supplementary Table 1 lists the genetic correlations between the male and female-specific GWAS, ranging from 0.56 for lean muscle volume index to 0.97 for muscle fat infiltration, as well as the locus yield.”

Supplementary Table 1. Results from sex-stratified GWAS on the body MRI measures, listing genetic correlations (r_g) between the male and female-specific GWAS, and locus yields.

Measure	r_g	Male yield	Female yield
Abdominal Fat Ratio	0.84	0	0
Abdominal Subcutaneous Adipose Tissue Volume index	0.79	1	1
Abdominal Subcutaneous Adipose Tissue Volume	0.73	1	1
Anterior Thigh Lean Muscle Volume index	0.56	0	0
Anterior Thigh Lean Muscle Volume	0.63	3	1
Anterior Thigh Muscle Fat Infiltration	0.89	6	6

Liver Proton Density Fat Fraction	0.8	6	3
Posterior Thigh Lean Muscle Volume index	0.58	0	0
Posterior Thigh Lean Muscle Volume	0.6	2	3
Posterior Thigh Muscle Fat Infiltration	0.97	7	11
Total Thigh Muscle Volume normalized	0.73	0	0
Visceral Adipose Tissue Volume index	0.76	0	0
Visceral Adipose Tissue Volume	0.78	0	0
Weight-to-Muscle Ratio	0.82	0	0

6. As they mention strength of the study, the authors are invited to provide a limitation section of their study, with mention of the overall lack of power to identify additional loci different from those covered by classical cardiometabolic traits.

Thank you for this suggestion. Besides the added text on the novelty of the loci, we agree that such a limitation section would give a good balance and highlight shortcomings to be resolved by future studies. We have now included such a section in the Discussion, as well as adjusted the concluding paragraph:

“Limitations of this study include the fact that the approaches employed do not allow for causal claims beyond genetic associations. Establishing the directionality of causal effects underlying the genetic correlations between the studied measures and diseases will therefore require follow-up investigation, e.g., through Mendelian randomization. Further, the limited locus yield from the univariate GWAS and the low percentage of explained genetic variance for the body MRI measures, with the exception of liver fat, point towards low statistical power. The collection of larger sample sizes, as now underway through several large-scale initiatives such as the UK Biobank, and the use of more powerful statistical approaches will be required to improve discovery.”

“However, our findings have made it clear that most of the body MRI measures are highly complex and polygenic, leading to limited yield with current sample sizes. Further, when combined, they tag similar sets of biological processes as widely available measures of anthropometrics and blood markers. This raises the question whether it is worthy to collect costly body MRI scans to obtain these measures, which is hard to answer firmly with current knowledge. This study does show that the individual measures have their own unique patterns of genetic correlations and that they lead to the identification of novel loci, indicating that they capture unique information, which may prove important to tease apart the influences of complex biological processes on body composition.”

Reviewer #3 (Remarks to the Author):

The manuscript describes briefly the genetic architecture of 31 traits extracted from large-scale MRI conducted in the UK Biobank. There are now several large studies looking at some of these and related traits (e.g. <https://www.nature.com/articles/s41591-019-0563-7> - there are many others) and investigating the contribution of these traits to cardiometabolic disease using genetic epidemiology. While several of the traits, especially those related to muscle composition, have not to my knowledge been studied at this scale before, I find the study overstates its novelty a little in this regard. A major finding is the genetic correlation between liver fat and type 2 diabetes pointing to a possible etiological role; this is also not a new hypothesis (e.g. <https://diabetesjournals.org/care/article-abstract/45/2/460/139118/Estimating-the-Effect-of-Liver-and-Pancreas-Volume>) The novel aspects in my opinion are: (1) definition of several traits and indices related to body composition; (2) integration of many adiposity and body composition traits into a single measure and conducting a multivariate analysis using software previously developed by the authors; (3) semi-systematic genetic correlation with metabolic risk factors and several disease outcomes, establishing and exploring the relationships between these traits.

1. Given the focus on MAFLD (is this the same as NAFLD?) in the introduction and discussion, it is somewhat surprising that this was not included in the genetic correlation analysis. How were the diseases of interest chosen?

MAFLD is a recent name change from NAFLD, reflecting an updated, more precise disease definition that promotes a better understanding and treatment, please see the cited paper by Eslam *et al.* for more information.²⁵ The reason for not including this condition in the genetic correlation analysis is simply because there is no well-powered GWAS data available. At most, several hundred cases have been studied, which would mean any genetic correlation estimates would be extremely noisy. We have now mentioned this explicitly in the newly added limitations section.

The diseases of interest were further chosen based on their prevalence and impact on society, as noted in the authoritative report on the global burden of non-communicable diseases that is cited in the introduction.¹⁶ Admittedly, this is a somewhat arbitrary selection, which we now also acknowledge in the limitations section:

“Our investigation of traits was further not comprehensive, and we lacked a sufficiently powered GWAS of MAFLD for inclusion in the analyses of genetic correlation.”

2. The discussion of the identified genes is very minimal. There have been several previously described studies in this cohort looking at related traits. How many of the loci identified in the univariate analysis are novel?

We acknowledge that the discussion of identified genes is minimal. This is a choice that follows from our main goal to provide an overview of the overall genetic architecture of multiple sets of measures, prioritizing the description of their overall characteristics over that of a selection of identified genes. We do however provide extensive overviews of these genes, and associated pathways, in the Supplementary Data, for those interested in specific associations.

Regarding locus novelty, we have now added Supplementary Figure 3 (please see above), which shows that the majority of the loci discovered for the body MRI measures were not significant for the cardiometabolic measures at comparable sample sizes. Please see the answer to the first comment of Reviewer #2 for this figure and for more information.

3. The methods do not adequately describe the generation of the MRI-derived measurements, and the reference does not give a complete picture either. What “manual adjustments” were performed? Were they performed for all 35,000 participants? The derivation of PDFF is also not adequately described and no reference is given here. A description of quality control processes (if any) is also missing.

We thank the reviewer for these questions. We agree that the generation of the MRI-derived measurements are not adequately described. In the online methods we have therefore added the following:

“The methods used to generate the MRI-derived measurements has been described and evaluated in more detail elsewhere.¹⁻⁶ Briefly, the process for fat and muscle compartments includes the following steps: (1) calibration of fat images using fat-referenced MRI, (2) registration of atlases with ground truth labels for fat and muscle compartments to the acquired MRI dataset to produce automatic segmentation, (3) quality control by two independent trained operators including the possibility to adjust and approve the final segmentation, and (4) quantification of fat volumes, muscle volumes and muscle fat infiltration within the segmented regions. For liver proton density fat fraction (PDFF), nine regions of interest (ROI) were manually placed, evenly distributed in the liver volume, while avoiding major vessels and bile ducts.

Muscle volumes were calculated as fat-tissue free muscle volumes. Muscle fat infiltration was calculated as the average T2*-corrected fat fraction and converted to proton density fat fraction.^{4,7} Liver PDFF was calculated depending on the protocols implemented by UK Biobank. The liver protocol was initially based on a single-slice symmetric 10-point acquisition with IDEAL reconstruction, but after 10.000 scans it was replaced by the whole-body dual echo Dixon images with an additional T2* and proton density correction, where the T2* values were estimated from a separate single-slice asymmetric 6-point acquisition. The latter method has been described and validated against IDEAL-based liver PDFF previously.⁴ 1.338 scans were acquired with both liver protocols to assess the switch, and a good agreement was found between the protocols - the mean difference in liver PDFF was 0.30% with a standard deviation of the differences of 0.80%.

Test-retest reliability of the MRI-derived measures included in this study is high, with a nearly perfect intraclass correlation coefficient, and the automated processing performs better than manual segmentation of muscle and fatty tissue.^{8,9} A recent study, investigating both regional fat and muscle volumes as well as muscle fat infiltration and liver fat fraction, showed high repeatability and reproducibility on five different 1.5T and 3T scanners from three different vendors.⁴”

4. On the genetics side, several sections are missing detailed methods: heritability estimation, genetic correlation methods, annotation of genes, etc. Also QC/sample inclusion criteria are not fully described - for example, were close relatives removed?

Our apologies for the lack of information on the application of these methods. We have now expanded the Methods as follows:

“We made use of the Functional Mapping and Annotation of GWAS (FUMA) online platform (<https://fuma.ctglab.nl/>) to map significant SNPs from the MOSTest analyses to genes. **For this, we combined the default positional mapping with eQTL and 3D chromatin interaction mapping, including all available tissue types.**”

“LD score regression

For estimates of SNP-based heritability (h_2), we applied LD score regression (LDSR)²⁶ to the univariate GWAS summary statistics. For this, each set of summary statistics underwent additional filtering, including the removal of all SNPs in the extended major histocompatibility complex region (chr6:25–35 Mb). We further used these munged summary statistics to perform cross-trait LDSR to estimate genetic correlations between the measures.²⁷”

In addition, please see the answer to the next point regarding our revised approach to dealing with close relatives in the sample.

5. A major finding of the study concerns the heritability and genetic architecture of the adiposity and muscle traits. The UK Biobank is known to oversample related individuals (<https://www.nature.com/articles/s41586-018-0579-z>) relative to their frequency in the general population. The method used does not account for this relatedness and is thus vulnerable to an inflation of type-1 error rate (presumably this would also affect the downstream analysis). I suggest that the authors check for residual confounding using one of the available standard methods (e.g. genomic control, LDSC regression intercept (<https://www.ncbi.nlm.nih.gov/pmc/articles/PMC6758917/>), or a QQ plot). In addition, correcting for covariates before rank normalization can re-introduce confounding (<https://www.nature.com/articles/s41431-018-0159-6>)

We indeed failed to consider participant relatedness in the original analyses, and we thank the reviewer for identifying this mistake. This has now been corrected by removing one of each pair of (second-degree or closer) related individuals, based on kinship coefficients as calculated through KING and released by the UK Biobank. We carried out this exclusion as the final step to minimize the number of pairs remaining. This is now described in the revised Methods section as follows:

“As a final step, we excluded one of each pair of related individuals in the remaining sample, as determined through KING¹³ and released by UKB, using a kinship coefficient threshold of 0.0884 (n=448), leaving N=33 588...”

“For the secondary analyses of measures of cardiometabolic health, we included all White Europeans with complete genetic and covariate data. After excluding one of each pair of related individuals (n=34,366), this sample consisted of 377,950 individuals...”

Regarding residual confounding, we followed our (admittedly undescribed) standard procedure of checking all the QQ plots and have found no evidence for this, in line with the extensive validation work we carried out for our GWAS pipeline, as described in Van der Meer *et al.* 2020.²⁸ Our phenotypes further showed low skew before normalization, as checked through histograms. We have further calculated the correlation between age and the normalized measures, which was negligible, ranging from -0.009 (for ATMFI) to 0.023 (for ASAT).

6. I don't see a data availability statement in the manuscript, although the box is checked. I encourage the authors to make their summary statistics available via GWAS catalog <https://www.ebi.ac.uk/gwas/>

We fully agree that data availability, encompassing the used software as well as GWAS summary statistics, is important to promote Open Science. The statement is present under 'Materials & Correspondence', following the Methods section, and includes a link to the mentioned catalog.

References

1. Linge, J. *et al.* Body composition profiling in the UK Biobank Imaging Study. *Obesity* **26**, 1785–1795 (2018).
2. West, J. *et al.* Feasibility of MR-based body composition analysis in large scale population studies. *PLoS One* **11**, e0163332 (2016).
3. Borga, M. *et al.* Advanced body composition assessment: from body mass index to body composition profiling. *J. Investig. Med.* **66**, 1–9 (2018).
4. Borga, M. *et al.* Reproducibility and repeatability of MRI-based body composition analysis. *Magn. Reson. Med.* **84**, 3146–3156 (2020).
5. Karlsson, A. *et al.* Automatic and quantitative assessment of regional muscle volume by multi-atlas segmentation using whole-body water–fat MRI. *J. Magn. Reson. Imaging* **41**, 1558–1569 (2015).
6. Leinhard, O. D. *et al.* Quantitative abdominal fat estimation using MRI. in *2008 19th International Conference on Pattern Recognition* 1–4 (IEEE, 2008).
7. Karlsson, A. *et al.* The effect on precision and T1 bias comparing two flip angles when estimating muscle fat infiltration using fat-referenced chemical shift-encoded imaging. *NMR Biomed.* **34**, e4581 (2021).
8. Thomas, M. S. *et al.* Test-retest reliability of automated whole body and compartmental muscle volume measurements on a wide bore 3T MR system. *Eur. Radiol.* **24**, 2279–2291 (2014).
9. Newman, D. *et al.* Test–retest reliability of rapid whole body and compartmental fat volume quantification on a widebore 3T MR system in normal-weight, overweight, and obese subjects. *J. Magn. Reson. imaging* **44**, 1464–1473 (2016).
10. Linge, J., Heymsfield, S. B. & Dahlqvist Leinhard, O. On the Definition of Sarcopenia in the Presence of Aging and Obesity—Initial Results from UK Biobank. *Journals Gerontol. Ser. A* **75**, 1309–1316 (2020).
11. Linge, J., Ekstedt, M. & Leinhard, O. D. Adverse muscle composition is linked to poor functional performance and metabolic comorbidities in NAFLD. *JHEP Reports* **3**, 100197 (2021).
12. Linge, J., Petersson, M., Forsgren, M. F., Sanyal, A. J. & Dahlqvist Leinhard, O. Adverse muscle composition predicts all-cause mortality in the UK Biobank imaging study. *J. Cachexia. Sarcopenia Muscle* **12**, 1513–1526 (2021).
13. Manichaikul, A. *et al.* Robust relationship inference in genome-wide association studies. *Bioinformatics* **26**, 2867–2873 (2010).
14. Gurholt, T. P. *et al.* Population-based body–brain mapping links brain morphology with anthropometrics and body composition. *Transl. Psychiatry* **11**, 1–12 (2021).
15. Firth, J. *et al.* The Lancet Psychiatry Commission: a blueprint for protecting physical health in people with mental illness. *The Lancet Psychiatry* **6**, 675–712 (2019).
16. Bloom, D. E. *et al.* *The global economic burden of noncommunicable diseases*. (Program on the Global Demography of Aging, 2012).
17. Alberti, K. G. M. M. *et al.* Harmonizing the metabolic syndrome: a joint interim statement of the international diabetes federation task force on epidemiology and prevention; national heart, lung, and blood institute; American heart association; world heart federation; international . *Circulation* **120**, 1640–1645 (2009).
18. Eckel, R. H., Alberti, K. G. M. M., Grundy, S. M. & Zimmet, P. Z. The metabolic syndrome. *Lancet* **375**, 181–183 (2010).
19. Shuster, A., Patlas, M., Pinthus, J. H. & Mourtzakis, M. The clinical importance of visceral adiposity: a critical review of methods for visceral adipose tissue analysis. *Br. J. Radiol.* **85**, 1–10 (2012).
20. Browning, L. M. *et al.* Measuring abdominal adipose tissue: comparison of simpler methods with MRI. *Obes. Facts* **4**, 9–15 (2011).
21. Beck, E., Esser, N., Paquot, N. & Scheen, A. J. Metabolically obese normal-weight individuals and metabolically healthy, but obese, subjects. *Rev. Med. Suisse* **5**, 1644–1646 (2009).
22. Britton, K. A. *et al.* Body fat distribution, incident cardiovascular disease, cancer, and all-cause mortality. *J. Am. Coll. Cardiol.* **62**, 921–925 (2013).

23. Stanley, M. R. I. A. S. G. of the L. A. R. G. G. D. dg108@ columbia. edu K. D. E. Y. J.-E. S. N. A. J. B. L. P.-S. F. X. H. Adipose tissue distribution is different in type 2 diabetes. *Am. J. Clin. Nutr.* **89**, 807–814 (2009).
24. Neeland, I. J. *et al.* Body fat distribution and incident cardiovascular disease in obese adults. *J. Am. Coll. Cardiol.* **65**, 2150–2151 (2015).
25. Eslam, M., Ratziu, V. & George, J. Yet more evidence that MAFLD is more than a name change. *J. Hepatol.* **74**, 977–979 (2021).
26. Bulik-Sullivan, B. K. *et al.* LD Score regression distinguishes confounding from polygenicity in genome-wide association studies. *Nat. Genet.* **47**, 291–295 (2015).
27. Bulik-Sullivan, B. *et al.* An atlas of genetic correlations across human diseases and traits. *Nat. Genet.* **47**, 1236 (2015).
28. van der Meer, D. *et al.* Understanding the genetic determinants of the brain with MOSTest. *Nat. Commun.* **11**, 1–9 (2020).

Reviewers' comments:

Reviewer #1 (Remarks to the Author):

Many thanks for revising your manuscript.

In the abstract:

The authors claim 'conducting the first genome-wide association study (GWAS) of these MRI-derived measures'. The GWAS of many of these traits has been published before (PMID: 34128465).

In the results:

Line 233 says 'Univariate GWASs on the individual measures revealed a total of 82 loci'; line 243 it says 'Of the 79 lead SNPs'. Please clarify why there is a mismatch.

Line 313, what is SI?

Figure legend for figure 2 is incomplete.

Please provide explanation on why you performed Multivariate test and how to interpret the results. What is the logic for which traits to include?

Lines 345-351, it is not clear to me why you limited the GWAS of cardiometabolic traits to 33,588 individuals if your aim was to see how many variants were novel? In the next paragraph, you have used the full set in the multivariate test. Please provide justification.

In the discussion:

Line 485, I am not sure how the Multivariate GWAS leads to this conclusion "The findings allow for numerous follow-up investigations; for instance, further studies are needed to clarify the role of putative moderators such as sex, age, and ethnicity." Please clarify.

Line 496, could you give an example for "more powerful statistical approaches"?

General:

I still did not find the supplementary figures, and the supplementary tables do not have legends.

Reviewer #2 (Remarks to the Author):

I thank the authors for addressing all my main points and I agree that the current version is substantially improved. I have no further comments.

Reviewer #3 (Remarks to the Author):

Thank you for the responses and the updates to the manuscript and methods. I have no further comments.

We are happy that our revisions were appreciated by the reviewers, and we wish to thank reviewer #1 for their remaining comments. Below, we provide a point-by-point response to these comments and indicate how we have updated the manuscript in accordance.

Reviewer #1 (Remarks to the Author):

Many thanks for revising your manuscript.

In the abstract:

The authors claim ‘conducting the first genome-wide association study (GWAS) of these MRI-derived measures’. The GWAS of many of these traits has been published before (PMID: 34128465).

Indeed, the listed study included three of the measures also investigated in the present study. The overall sets of measures studied did differ substantially, with a focus on organ tissue traits by Liu *et al.* versus our focus on body composition, as stated in the first part of the quoted sentence. Nonetheless, we agree that this statement can be misconstrued and we have therefore altered the statement as follows:

“Here, we aimed to elucidate the genetic architecture of body composition, by conducting genome-wide association studies (GWAS) of these MRI-derived measures.”

In the results:

Line 233 says ‘Univariate GWASs on the individual measures revealed a total of 82 loci’; line 243 it says ‘Of the 79 lead SNPs’. Please clarify why there is a mismatch.

Please note that the sentence reads “Of the 79 lead SNPs available in this set,”. Preprocessing of genetic data includes filters for e.g. Hardy-Weinberg equilibrium and excessive missingness, which will vary depending on sample composition, explaining why a small subset of SNPs are not available in the additional (non-White European) sample. These processing steps are listed in the Methods section, and we believe that listing them in the Results section is not favorable for the manuscript.

Line 313, what is SI?

SI stands for Supplementary Information. This abbreviation has been introduced in the second paragraph of the Results section. The SI contains the supplementary figures and legends.

Figure legend for figure 2 is incomplete.

Thank you for pointing this out. It appears indeed that the textbox extended beyond the bottom of the page, making the text illegible in the pdf file. This has now been corrected.

Please provide explanation on why you performed Multivariate test and how to interpret the results. What is the logic for which traits to include?

A multivariate test enables us to jointly analyze all measures of interest, taking advantage of overlapping effects to boost statistical power and discover shared determinants. How to interpret the results is guided by what is tagged by the set of measures. In this case, the identified genetic variants can therefore be interpreted as contributing to body composition as a whole. The logic for which traits to include is in line with this, we aimed to investigate body composition, and therefore included traits that contribute to body composition. We have now revised the first sentence of the paragraph on the multivariate analyses:

“Gene variants are likely to have distributed effects across these measures of body composition, as they are correlated components of the same biological system. To identify variants influencing body composition as a whole, we also jointly analyzed all measures through the Multivariate Omnibus

Statistical Test (MOSTest),⁴⁹ which increases statistical power in a scenario of shared genetic signal across the univariate measures.⁴⁹⁻⁵¹

Lines 345-351, it is not clear to me why you limited the GWAS of cardiometabolic traits to 33,588 individuals if your aim was to see how many variants were novel? In the next paragraph, you have used the full set in the multivariate test. Please provide justification.

This approach was taken following the request of another reviewer to indicate whether our analyses allowed for the identification of novel loci, compared to classical biochemical measures, at comparable sample sizes. This came from a general interest in knowing whether these body MRI measures truly allow for more discovery, without being confounded by differences in sample sizes. We have now clarified this in the manuscript:

“To ascertain whether the body MRI measures truly allow for more discovery, without being confounded by differences in sample size, we restricted these analyses to the same sample of individuals with available MRI data (N=33,588). These analyses showed that the large majority of loci were indeed novel;”

In the discussion:

Line 485, I am not sure how the Multivariate GWAS leads to this conclusion “The findings allow for numerous follow-up investigations; for instance, further studies are needed to clarify the role of putative moderators such as sex,⁷⁶ age, and ethnicity.” Please clarify.

We apologize for this poor phrasing. These specific findings indeed do not explicitly enable those follow-ups. Rather, we simply meant that the MOSTest findings will leave room for further influences to be investigated. This is now rephrased as follows:

“However, further studies are needed to clarify the role of putative moderators such as sex,⁷⁶ age, and ethnicity.⁷⁷”

Line 496, could you give an example for “more powerful statistical approaches”?

The applied MOSTest approach is such an example. Other approaches could be those that make use of Bayesian statistics to integrate information from a secondary trait to boost the identification of loci in a primary trait of interest, e.g. conditional false discovery rate analysis. We have now added the suggestion of using MOSTest to this sentence:

“and the use of more powerful statistical approaches, such as MOSTest, will be required to improve discovery.”

General:

I still did not find the supplementary figures, and the supplementary tables do not have legends.

We are sorry to read that this was still the case. Please see below a screenshot of the files that were included with revision #1, with the supplementary information listed at number 6. We have downloaded and double-checked this file, it is complete, containing the figures and description of tables.

1. Author Cover Letter PDF (76KB)
2. "Rebuttal Letter" PDF (226KB)
3. Article File PDF (641KB)
4. Updated editorial policy checklist PDF (1479KB)
5. Updated reporting summary PDF (1461KB)
6. Supplementary information PDF (712KB)
7. Supplementary Data 1 PDF (42KB)
8. Supplementary Data 2 PDF (50KB)
9. Supplementary Data 3 PDF (145KB)
10. Supplementary Data 4 PDF (38KB)
11. Supplementary Data 5 PDF (137KB)
12. Supplementary Data 6 PDF (86KB)
13. Reviewer Zip File "Zip of files for Reviewer"

REVIEWERS' COMMENTS:

Reviewer #1 (Remarks to the Author):

Thank you for the responses and the updates to the manuscript and methods. I have no further comments.